# Data-driven protease engineering by DNA-recording and epistasis-aware machine learning

Lukas Huber [1,6], Tim Kucera [1,2,3,6], Simon Höllerer [1], Karsten Borgwardt [1,2,3] ✉, Sven Panke [1] ✉ & Markus Jeschek [1,4,5] ✉

Protein engineering has recently seen tremendous transformation due to machine learning (ML) tools that predict structure from sequence at unprecedented precision. Predicting catalytic activity, however, remains challenging, restricting our capabilities to design protein sequences with desired catalytic function in silico. This predicament is mainly rooted in a lack of experimental methods capable of recording sequence-activity data in quantities sufficient for data-intensive ML techniques, and the inefficiency of searches in the enormous sequence spaces inherent to proteins. Herein, we address both limitations in the context of engineering proteases with tailored substrate specificity. We introduce a DNA recorder for deep specificity profiling of proteases in *Escherichia coli* as we demonstrate testing 29,716 candidate proteases against up to 134 substrates in parallel. The resulting sequence-activity data on approximately 600,000 protease-substrate pairs does not only reveal key sequence determinants governing protease specificity, but allows to build a data-efficient deep learning model that accurately predicts protease sequences with desired on- and off-target activities. Moreover, we present epistasis-aware training set design as a generalizable strategy to streamline searches within enormous sequence spaces, which strongly increases model accuracy at given experimental efforts and is thus likely to have implications for protein engineering far beyond proteases.

Proteases play critical roles in various biological processes, including gene regulation, signal transduction and protein homeostasis. Their ability to cleave peptide bonds is the basis for their use as research reagents, detergent additives and food processing agents[1]. Moreover, they are important drug targets, and proteases themselves are therapeutically used, for instance, as drug-releasing or wound-healing agents[2]. Perhaps most intriguingly, proteases could be used to directly target disease-related proteins, a promising yet so far largely underexplored concept[1–3]. Recent developments in this context show potential for treating cancer, asthma, or neuroendocrine disorders, and in regenerative medicine[4–6]. In this context, high substrate specificity is crucial since cleavage of off-target proteins in a patient may cause undesirable side effects[2]. Notably, commercial proteases still mainly rely on native specificities, which are limited to a few cleavable amino acid motifs unlikely to occur in any target protein of interest. Accordingly, the ability to re-engineer protease specificity would

¹Department of Biosystems Science and Engineering, ETH Zurich, Basel, Switzerland. ²Swiss Institute of Bioinformatics, Basel, Switzerland. ³Department of Machine Learning and Systems Biology, Max Planck Institute of Biochemistry, Martinsried, Germany. ⁴Synthetic Microbiology, University of Regensburg, Regensburg, Germany. ⁵Laboratory of Synthetic and Applied Microbiology, SB ISIC & SV IBI, École Polytechnique Fédérale de Lausanne (EPFL), Lausanne, Switzerland. ⁶These authors contributed equally: Lukas Huber, Tim Kucera. ✉e-mail: borgwardt@biochem.mpg.de; sven.panke@bsse.ethz.ch; markus.jeschek@epfl.ch

substantially widen their pharmaceutical scope and have a transformative impact across the life sciences[1,3,7,8].

Following this potential, several high-throughput (HTP) approaches for directed protease evolution have been developed, in which proteolytic cleavage is linked either to fluorescent or selectable phenotypes[1,3,9–32], most notably recent efforts relying on proteolysis-dependent phage propagation[4,33]. Crucially, available HTP approaches establish a link between protease sequence and proteolytic activity for a few hits only, while such sequence-activity data are not collected for most members of the library. This prohibits deeper insights into sequence determinants that dictate protease activity and specificity. Consequently, new screens must be established and carried out for each new protease or target at hand. Importantly, large-scale sequence-activity data on proteases, if available, could be exploited to train machine-learning (ML) models that generalize across large sequence-activity landscapes to accelerate protease development significantly as we and others have previously shown for different enzymes[34–39].

A major drawback of available HTP approaches for protease engineering is the difficulty to systematically integrate off-target activity during screening or selection. Typically, variants are initially screened for activity towards one desired target, and subsequently, only a few hits are tested more extensively for off-target activity. However, evolution tends to progress through generalist enzyme stages before acquiring high specificity[40], which often leads to increased promiscuity[33,41–47]. While often unproblematic or even desired, promiscuity can be detrimental for therapeutic proteases due to potential side effects. Thus, retroactive off-target testing is highly prone to unsuccessful outcomes, and approaches that quantify off-target activity already during the initial HTP step are highly desirable. To avoid the emergence of off-target proteolysis, the use of counterselection substrates, mainly the protease's native substrate, has been suggested[4,9,15,16,23–25,32,48], which so far has been limited to at most two off-targets[26]. Notably, amongst the numerous studies reporting on protease specificity engineering, only a few involve systematic substrate profiling[17,23,33,48], which remains difficult to integrate with available approaches.

To overcome these limitations, we present herein a recombinase-based DNA recorder for proteolytic activity that features HTP collection of sequence-activity data on proteases tested against large numbers of potential substrates simultaneously (i.e., specificity profiling). Relying solely on next-generation sequencing (NGS) as experimental readout, the recorder enables kinetic activity measurements on hundreds of thousands of protease-substrate combinations while concomitantly testing for off-target activity in a single experiment. Further, we use the recorded data on approximately 600,000 protease-substrate pairs by training ML models capable of finding variants with desired specificity profiles within a combinatorial space of 64 million protease sequences, and introduce a novel sampling strategy based on epistatic priors (i.e., "epistasis-aware" ML) to maximize data efficiency and model performance at given experimental effort.

## Results

### Recording proteolytic activity within DNA

To record proteolytic activity in DNA, we built upon our previous experience with DNA recorders for HTP assessment of gene regulatory elements[49,50]. Precisely, we sought to build a genetic device that encodes three types of information in a single, short DNA molecule: (i) the sequence of the candidate protease; (ii) the sequence of the candidate substrate; and (iii) the proteolytic activity of the underlying protease-substrate pair. Crucially, we anticipated that current NGS technology would allow to read these information (i–iii) for millions of protease-substrate pairs in parallel as previously shown for ribosome binding sites (RBSs) and coding sequences (CDSs)[49,50]. As a model protease, we selected Tobacco Etch Virus protease (TEVp) with its

substrate (TEVs, canonical motif: ENLYFQ|S) due to its high substrate specificity, facile expression, and functionality across pro- and eukaryotes[51]. We conceived a plasmid architecture containing expression cassettes for TEVp and the phage recombinase Bxb1 (Fig. 1a). Bxb1 is fused to a C-terminal peptide containing TEVs followed by a proteolytic degradation signal (SsrA)[52]. Moreover, the architecture entails a recombination array flanked by Bxb1's attachment sites.

Briefly, the architecture works as follows: In the absence of TEVp activity, Bxb1 undergoes fast intracellular degradation due to the SsrA signal, and thus recombination activity is low. If, however, a TEVp variant cleaves TEVs, it releases SsrA and thus stabilizes Bxb1, leading to accelerated modification of the recombination array from its initial (unflipped) to an inverted (flipped) state. Crucially, the fraction of flipped recombination arrays (fraction flipped) amongst all copies of the architecture for a given protease-substrate combination should directly correlate with the proteolytic activity of this pair. Using NGS, this fraction flipped can be determined and tracked over time for extremely large numbers of variants in parallel, as previously shown[49,50]. Lastly, TEVp- and TEVs-specific DNA barcodes are used to retrieve the sequences of TEVp and TEVs candidates for each NGS read, thus establishing a direct link between each protease-substrate pair and corresponding proteolytic activity (Fig. 1b, "**Methods**").

Following this architecture, we constructed several plasmids employing different TEVp variants previously engineered in our group[53] paired with the canonical TEVs (Supplementary Fig. S1a). Relying on fluorescent labeling ("**Methods**")[49], we tested whether TEVp activity would indeed lead to stabilization of Bxb1. This was confirmed by a marked increase in cellular Bxb1-levels upon TEVp expression compared to catalytically inactive and protease-free controls (Supplementary Fig. S1b). Further, we optimized the accessibility of TEVs by adding flexible amino acid linkers flanking the protease substrate, which yielded improved signal-to-background ratios (Supplementary Fig. S1b). Next, we examined whether TEVp-mediated proteolysis also translates into increased recombination activity. We adapted our previous DNA recorder workflow to the case of proteases at hand[49] ("**Methods**"): Briefly, an E. coli strain carrying the plasmid-encoded DNA recorder is grown in a shake flask, and samples are drawn at different times after Bxb1 induction. After plasmid extraction, target fragments for NGS containing TEVp- and TEVs-specific barcodes and the recombination array are isolated via a PCR-free protocol (Fig. 1b). Fragments are ligated to DNA duplex adapters with sample-specific indices for Illumina NGS, pooled and collectively sequenced. Finally, NGS data are processed to obtain the sequences of the TEVp (i) and TEVs (ii) candidates, and the fraction flipped (iii) over time (flipping curve). Note that TEVp and TEVs sequences are separately obtained via long-read NGS (PacBio) to assign each variant's sequence to the specific barcodes in lookup tables ("**Methods**"). These barcodes are read in lieu of TEVp and TEVs sequences in Illumina NGS and replaced by the actual amino acid sequences during data processing relying on the lookup tables.

Following this protocol, we first optimized the timing of *bxb1* expression (induction time point) as well as the ratio between TEVp and Bxb1 by promoter and RBS engineering, respectively (Supplementary Fig. S2). Applying the optimized DNA recorder setup, we then tested different TEVp variants against canonical TEVs (Fig. 1c). Active TEVp variants consistently showed elevated recombination activity compared to catalytically inactive counterparts (C151A mutants). Moreover, differential activity between variants was clearly distinguishable, indicating successful recording of proteolytic activity in DNA. The C151A mutants exhibited smaller signal increases over a control lacking TEVp, which was also reflected during the initial linker optimization (Supplementary Fig. S1b). In a previous study, similar observations were attributed to mere binding without cleavage by inactive TEVp[20], which in our case seems to cause a minor degree of

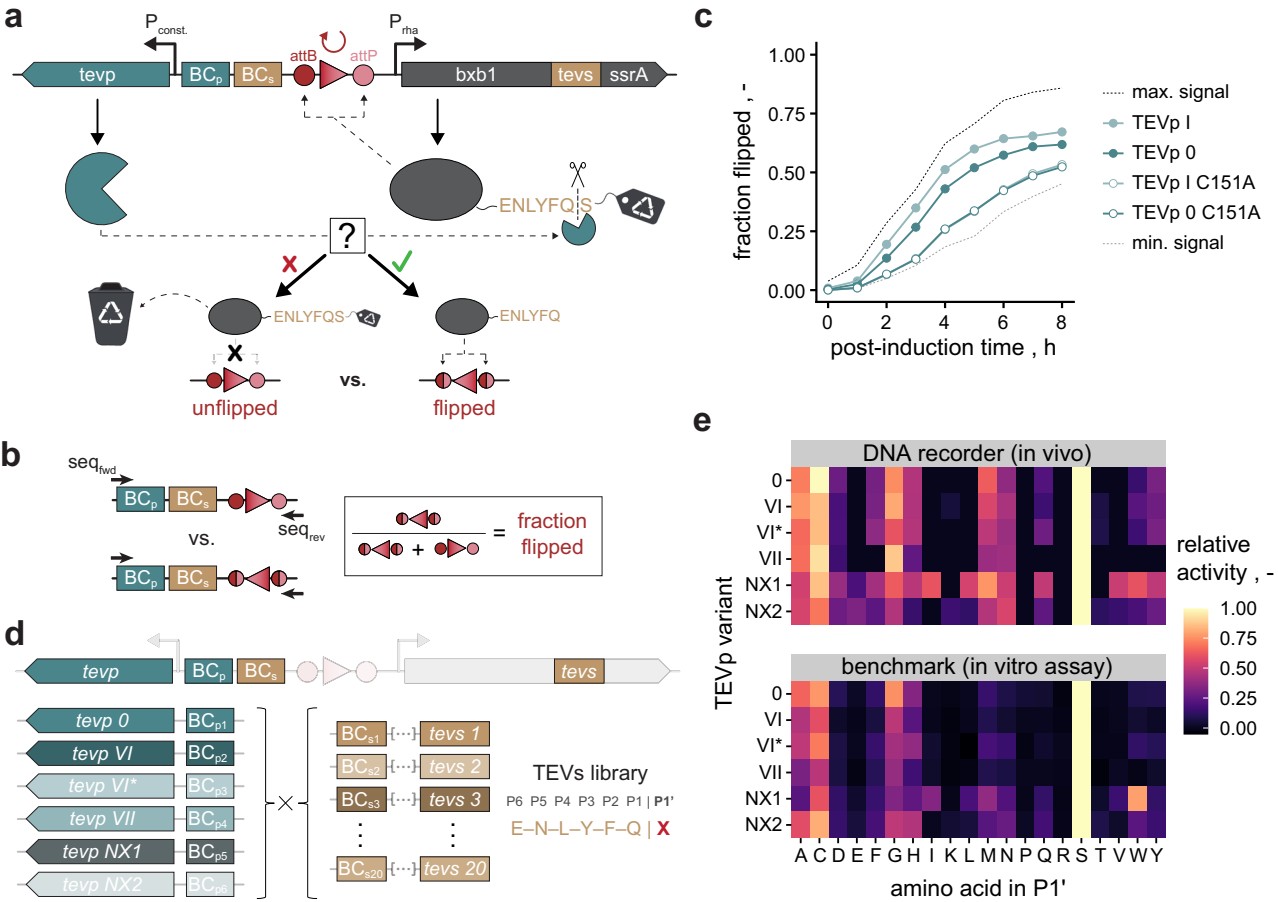

**Fig. 1 | DNA recorder for proteolytic activity. a** Genetic architecture containing *tevp* and *bxb1* genes controlled by a constitutive (P_const.) and a rhamnose-inducible (P_rha) promoter[81], respectively. A recombination module (red) flanked by Bxb1's attachment sites (attB/P) as well as TEVp- and TEVs-specific barcodes (BC_p/s) are included. *Bxb1* is fused to a degradation signal (*ssrA*) via a TEVs-containing linker. In the absence of TEVs cleavage, Bxb1 is rapidly degraded due to the degradation signal, and thus, recombination activity is low. If a candidate protease cleaves TEVs, Bxb1 is rescued from degradation, leading to faster recombination. **b** Target fragments for paired-end NGS. Using forward and reverse sequencing primers (seq_fwd/rev), both the identity/sequence of the protease-substrate pair (via BC_p/s) and the state (unflipped vs. flipped) of the recombination module can be read in high throughput. Reading of multiple copies of the target fragment allows determination of the fraction of flipped recombination modules for each protease-substrate pair as a quantitative metric for proteolytic activity. **c** Flipping curves (*n* = 1) obtained via the optimized DNA recorder

for proteolytic activity. Two differentially active variants (TEVp 0/I) were tested against canonical TEVs (ENLYFQ | S) along with corresponding catalytically inactive derivatives (C151A mutants). To mimic maximum possible proteolytic activity (max. signal), a control with an SsrA-less (i.e., fully stabilized) Bxb1 was included. As minimal signal (i.e., no TEVs cleavage), a control expressing mCherry instead of TEVp was included. **d** Proof of concept combining six TEVp variants with all 20 P1'-variants of TEVs (i.e., ENLYFQ | X library). TEVp variants 0, VI, VI* and VII have a similar P1'-specificity as the wildtype protease (i.e., preference for A, C, G or S), whereas variants NX1 and NX2 were engineered towards relaxed P1'-specificity[53]. **e** P1'-specificity profiles of the TEVp variants from (d) as determined in vivo using the DNA recorder (top) or in individual in vitro assays (bottom). Displayed activities were normalized to the activity on the canonical TEVs motif. TEVp variants and designations are derived from a previous study[53]. Source data are provided as a Source Data file.

substrate stabilization. This is corroborated by the fact that the binding contribution to the recorder's readout, unlike actual proteolysis, remains constant over time (Supplementary Fig. S3a). Importantly, it is relatively low compared to the signal for proteolysis (Supplementary Fig. S3a, **b**) and can be accounted for applying a minimal threshold above which variants are considered catalytically active (see below).

Next, we benchmarked the DNA recorder against an in vitro assays testing different TEVp variants individually on all 20 P1'-variants of TEVs (Fig. 1d, "**Methods**"). Four TEVp variants with similar P1'-specificity as the wild-type protease (i.e., preference for A, C, G or S) and two with relaxed P1'-specificity (i.e., acceptance of more amino acids) were tested[53]. As a DNA recorder-derived metric for proteolytic activity, we selected the area under the flipping curve within four hours post induction (AUC_4h), since this parameter exhibited the best signal amongst tested intervals and excellent robustness across replicate cultivations (Supplementary Fig. S4). To account for the aforementioned binding contribution and ensure robustness against experimental noise, we subtracted from the AUC_4h the mean value of

different, proteolytically inactive controls and regarded variants within a 95% confidence interval of these controls as inactive (i.e., value set to zero, Fig. S5). The resulting noise-adjusted value successfully recapitulated the in vitro measurements and thus was used as a DNA recorder metric for proteolytic activity hereafter (Fig. 1e). More specifically, all tested TEVp variants showed highly similar P1'-specificity profiles in both assays with strong correlation (Fig. 1e and Supplementary Fig. S6). This confirms the validity of the DNA recorder for proteolytic activity and its capability to test several protease variants on multiple substrates in parallel.

**Two-dimensional mutational scan of protease-substrate pairs**
Next, we capitalized on the DNA recorder's features to systematically assess the mutability of TEVp, in particular the question which residues tolerate mutations or dictate specificity for different substrate positions. We generated a library combining all 4446 theoretically possible TEVp single-site mutants with all 133 possible single-site substituents of canonical TEVs (Fig. 2a, "**Methods**"). As parent, we chose TEVp 0, a

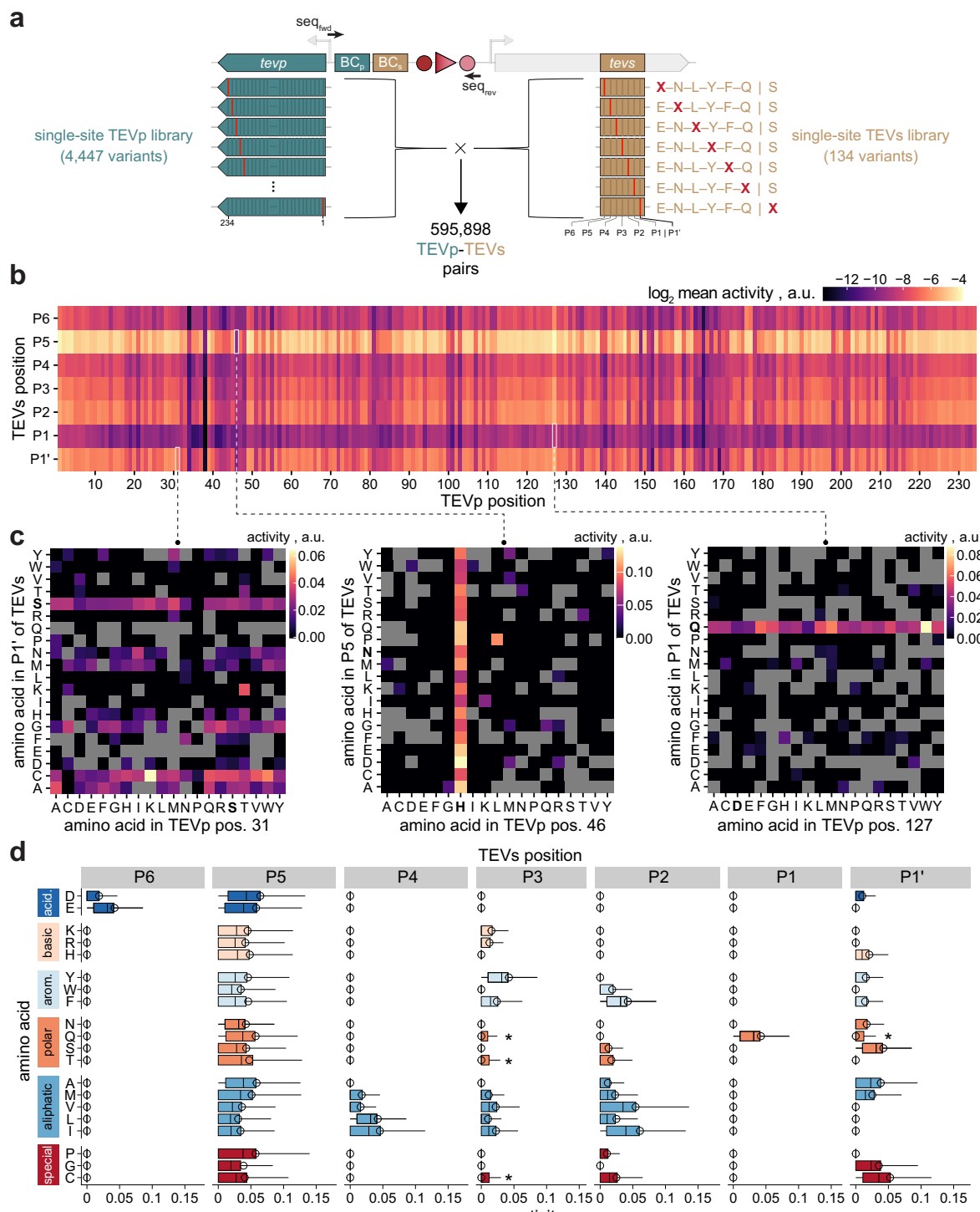

**Fig. 2 | Two-dimensional mutational scan of TEVp-TEVs pairs. a** A double-barcoded library combining all single-site substitution mutants of TEVp (i.e., 234 residues × 19 amino acids = 4446 variants plus parent TEVp 0) with all single-site substituents of TEVs (i.e., 7 residues × 19 amino acids = 133 variants plus canonical substrate) was generated, yielding a theoretical size of 595,898 TEVp-TEVs combinations. **b** Overview of proteolytic activities for 355,097 unique TEVp-TEVs pairs from the library in (**a**) above, high-quality read count threshold as determined by the DNA recording approach (Methods). The log2 of the mean activity across the mutated TEVp and TEVs positions (i.e., up to 400 TEVp-TEVs pairs per rectangle) is displayed. **c** Exemplary heatmaps detailing data underlying the rectangles with a white outline in (**b**). Parent amino acids are highlighted in bold face for TEVp and TEVs. Gray boxes indicate missing data. Note that the activity metric is identical to (**b**) but displayed without log2-transformation. **d** Specificity profile of the TEVp

library shown in (**a**) on all single-site mutants of the canonical TEVs. Boxplots show median activities of all library members on single TEVs variants bearing the indicated amino acid in the respective position, while for the respective other TEVs positions, the canonical amino acids are maintained. Amino acids are grouped by chemical category and ranked by hydrophobicity within each category. The canonical TEVs motif ENLYFQS is highlighted by thick box outlines. Boxes outline interquartile range with median (center line) and 20/80 percentiles (whiskers). Circles depict activities of the parent variant TEVp 0. Asterisks highlight acceptance of Q, T, and C at P3 as well as Q at P1′, which are not accepted by the parent. The number n of variants per boxplot is variable (average = 2715) and can be found in the source data underlying this figure. Source data are provided as a Source Data file.

mutant with increased solubility and stability compared to the wild-type protease[53]. The library was assembled from barcoded half-libraries of protease and substrate variants. Transformation of *E. coli* with the combined library yielded approximately 2,000,000 individual colonies. We then tested this library with the DNA recorder workflow, including inactive controls for noise correction and the six TEVp variants used above (Supplementary Fig. S7). The latter again exhibited a good agreement between DNA recorder and in vitro activity, in this case across all seven TEVs sites. Applying a threshold of ≥20 reads per sampling time point and barcode pair yielded activity data on approximately 355,000 unique TEVp-TEVs combinations (59.7% of possible combinations). 4437 (99.8%) and 134 (100%) of theoretically possible TEVp and TEVs variants were represented, respectively, with an average of about 80 substrates per TEVp variant (for details see Supplementary Fig. S8).

We first investigated the subset of data obtained for canonical TEVs. Mutating catalytic-triad residues expectedly resulted in complete inactivation, which was also observable for several other residues (Supplementary Fig. S9a). By contrast, other regions were amenable for mutation barely affecting activity, most notably a surface-exposed, mostly unstructured region distant from the substrate groove (residues 113-128) and the C-terminus from residue 219 onwards (Supplementary Fig. S9b, c). Truncating TEVp after this residue is reportedly tolerated[54], which stop codon mutants in our library confirmed (Supplementary Fig. S9d). Furthermore, several regions showed potential for activity improvement, including the N-terminus, where positive effects can likely be attributed to expression level changes resulting from CDS mutation[50]. Interestingly, while mutation of substrate-interacting residues[55] was expectedly prone to activity loss, our data show that N176 and W211 are exceptions to that end (Supplementary Fig. S9b and Supplementary Table S1). Mutation of N176 is generally well tolerated in line with previous reports, where substitutions to D, I, R or T had emerged during directed TEVp evolution[9,18,33,53]. Similarly, W211 mutants reportedly increase acceptance of Y (instead of F) at P2[29], which our data support, possibly due to a widened substrate pocket upon replacement of the bulky W211. A detailed analysis on the positional impact of TEVp mutation on activity towards canonical TEVs is provided in Tables S2 and S3, which may be consulted for future engineering efforts.

To explore potential routes to modulate substrate specificity, we next analyzed the entire dataset of the two-dimensional scan (Fig. 2b and Supplementary Fig. S10). Figure 2c illustrates exemplary cases of complete mutational inflexibility for TEVp with high flexibility for mutation in TEVs and vice versa, as well as a case with strong indications for epistasis between TEVp and TEVs residues. Since such anecdotal examples or positional averages do not deliver generalizable information for the engineering of specificity, we further performed more systematic analyses. We investigated the activity profile of all library members across all 134 TEVs variants (Fig. 2d and Supplementary Fig. S11). Expectedly, the library globally followed substrate preferences of the parent TEVp 0 (Supplementary Fig. S12), with, for instance, strong preferences for acidic or aliphatic amino acids in P6 or P4, respectively. P1 showed a strong preference for Q, whereas the other TEVs positions showed higher (P3, P2, P1') or even complete (P5) mutational flexibility in line with previous findings for wildtype-like TEVp variants[21,33]. A closer inspection of Fig. 2d unveiled potential avenues to modulate specificity, most notably for P3 and P1'. To this end, various candidates appear to accept Q, T, and C in P3 or Q in P1', which are not accepted by the parent.

A crucial risk when consulting positional means alone is that potentially interesting individual combinations of mutations between TEVp and TEVs are missed due to averaging out of effects. Striking examples are TEVp mutants N171D and N176T, which reportedly change P6-specificity to accept P and T, respectively[9]. While masked in positional analyses (Fig. 2d), our data clearly support these findings upon individual inspection (Fig. 3a). Furthermore, they hint to alternative TEVp substitutions lending novel P6-specificity. Prompted by

this example, we searched for target residues with high potential to change specificity in substrate positions P5 and P1. These were selected since the wildtype protease is virtually unspecific for P5 while accepting only Q in P1. Specifically, we searched for target residues that increase or decrease specificity for P5 or P1, respectively ("**Methods**"). Multiple substituents for residues 38, 81, 101 and 152 show diminished activity for most P5-variants narrowing down cleavage to one or few amino acids indicating arising P5-specificity (Fig. 3b). This highlights the potential of the two-dimensional approach, since these variants are representative of a reduced off-target activity and would not have been identified in conventional one-substrate screen. By contrast, widening P1-specificity strongly depends on which amino acids should be accepted, as reflected by different top-ranking target residues in each case (Supplementary Table S4). We thus computed a metric for promiscuity and ranked TEVp variants accordingly (Supplementary Fig. S13). Generally, no variants were found with significant activity on more than four amino acids in P1, indicating that it may not be possible to achieve pronounced promiscuity with a single mutation. Nonetheless, several variants showed novel P1-specificities, most notably variant P92M, which accepts D, F and L at comparable levels as the canonical Q, and to a lesser degree also K, M, R and S (Fig. 3c). For the other TEVs positions, P92M largely resembles substrate preferences of TEVp 0 (Supplementary Fig. S12).

## Multi-site mutagenesis and ML-guided engineering of protease specificity

While such single-site mutational scans provide useful insights and potential target residues, they disregard epistasis between different residues due to their position-wise approach. Considering epistasis is, however, essential to genuinely re-engineer protease specificity, imposing a need for multi-site mutagenesis. To examine the potential of our DNA recorder in this regard, we selected P1'-specificity as a testbed. Unlike other TEVs positions, P1' is surface exposed in the substrate-bound enzyme (Fig. 4a)[55], leading to certain promiscuity with a preference for small amino acids (A, C, G, S) as previously described[33,56] and confirmed herein (Supplementary Fig. S12). To assess the malleability of P1'-specificity, we selected six TEVp residues in two clusters. Cluster A (T30, S31, L32) is within a 3-Å radius of P1' on one side of TEVs, whereas cluster B (F217, M218, V219) lies approximately 5 Å on the opposite side (Fig. 4a). Crucially, the mutational scan had indicated that these residues are highly impactful for P1'-specificity (Supplementary Fig. S14). We generated three libraries targeting residues from either cluster (libraries A and B) or both clusters simultaneously (library AB) using NNK or NNS codons, and combined them with all 20 P1'-variants (Fig. 4b, "**Methods**"). Our DNA recorder pipeline yielded data on 61,185, 14,156, and 18,8717 unique TEVp-TEVs combinations for libraries A, B and AB, respectively, with uniform amino acid distributions at all randomized residues (Supplementary Fig. S15). Figure 4c displays P1'-specificity profiles of all members of the three libraries. As expected, a large fraction of candidates completely lost activity, which was more pronounced for library AB with six instead of three randomized residues. Amongst active candidates, many exhibited preferences similar to wildtype TEVp, and various others showed distinct specificity profiles. From the latter, we selected candidates with altered specificity or higher promiscuity for P1' from library A for re-testing at full substrate coverage. This substantiated the modulation of P1'-preferences, most notably variant T30G + L32T accepting W but rejecting canonical S, and three promiscuous variants (T30A, T30A + S31T, T30D; Fig. 4d). Purification of these variants confirmed our findings in vitro (Fig. 4e, "**Methods**").

However, increasing the number of target residues rapidly leads to sequence search spaces of unscreenable size. Accordingly, despite the high throughput of the DNA recorder, we only covered 0.028% of the $20^6$ theoretically possible TEVp variants and 0.015% of the $20^7$ possible TEVp-TEVs combinations for library AB. Thus, we sought to use ML to

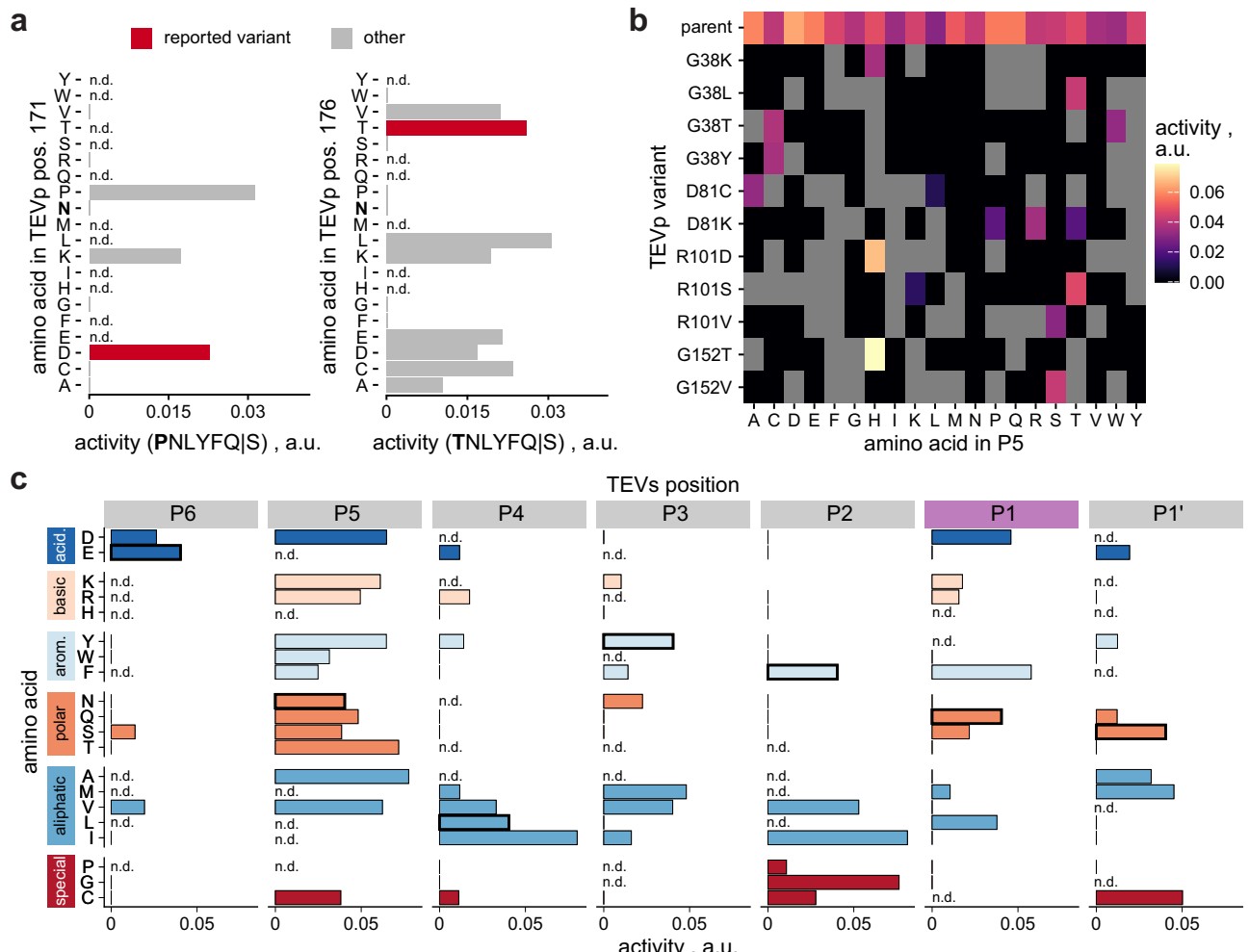

**Fig. 3 | Single-site mutants of TEVp with modulated substrate specificity.**
**a** Impact of substitution of TEVp residues 171 (left) and 176 (right) on the activity towards TEVs variants with P (left) and T (right) in P6 of TEVs. TEVp variants N171D and N176T were previously reported to accept P and T at P6, respectively[9]. Missing data points are indicated (n.d.). **b** TEVp variants with novel specificity at P5. For comparison, the parent variant TEVp 0 of the library is display on top. **c** Specificity profile of TEVp variant P92M on single-site mutants of canonical TEVs. The variant

was selected due to its relaxed substrate specificity for P1. Bars show activities on TEVs variants bearing the indicated amino acid in the respective position, while for the respective other positions, the canonical amino acids are maintained. Amino acids are grouped by chemical category and ranked by hydrophobicity within each category. The canonical TEVs motif ENLYFQS is highlighted by thick bar outlines. Missing data points are indicated (n.d.). Source data are provided as a Source Data file.

generalize from subsample data and provide an accurate model of the sequence-activity landscape and enable protease design in silico. We chose Multilayer Perceptrons (MLPs) because of their ability to model non-linear interactions (epistasis) at comparably low model complexities, facilitating rapid prediction (Fig. 4f). We framed the model as a multi-output classifier of binary activities on different substrates, accounting for class imbalance resulting from the high fraction of inactive TEVp-TEVs pairs (Fig. 4c). This also enables straightforward definition of probabilistic specificity profiles indicating activity or inactivity of a given TEVp variant on multiple on- and off-targets of interest. Other architectural details were optimized in a hyperparameter search, and the best model was determined on a held-out validation set ("**Methods**"). We trained the MLP and a diverse set of other ML models of varying complexity with increasingly large fractions of data from library AB, and evaluated performance on a held-out test set of 1000 TEVp variants using different metrics (Fig. 4g and Supplementary Fig. S16). The MLP revealed a stronger performance than the other models across all dataset sizes and performance metrics, most notably an area under the precision-recall curve (AUPRC) of 0.59 for the largest training set of 10,000 TEVp variants. The MLP distinctly outperforms classical ML techniques, likely due to its ability to model highly non-linear

dependencies. The high throughput of our DNA recorder is imperative for high prediction accuracy, as model performance does not plateau even for the largest training set in this study. Notably, combining the MLP with the large language model ESM[57] led to similar performance while strongly adding to computational expense. Using the MLP alone allowed us to predict the entire sequence-activity landscape of cluster AB ($20^7$ TEVp-TEVs pairs) within minutes on a single 10GB GPU, which is practically infeasible with existing language models. Crucially, this allows to identify candidate sequences predicted to be globally optimal with respect to desired specificity profiles. To this end, we searched the aforementioned test set for TEVp variants that would accept (on-target) or reject (off-target) different amino acids in P1', and compared hit rates of random versus MLP-guided screens (Fig. 4h). Remarkably, MLP-guided design drastically increased screening efficiency as reflected by mean and maximum improvements of the hit rate of 5- and 48-fold for on-target and 8- and 13-fold for off-target profiles, respectively.

**Boosting ML performance with epistatic priors**
Lastly, we aimed to further increase the efficiency of our ML-guided engineering approach by optimizing experimental data acquisition through training set design[58,59]. For instance, we have recently shown

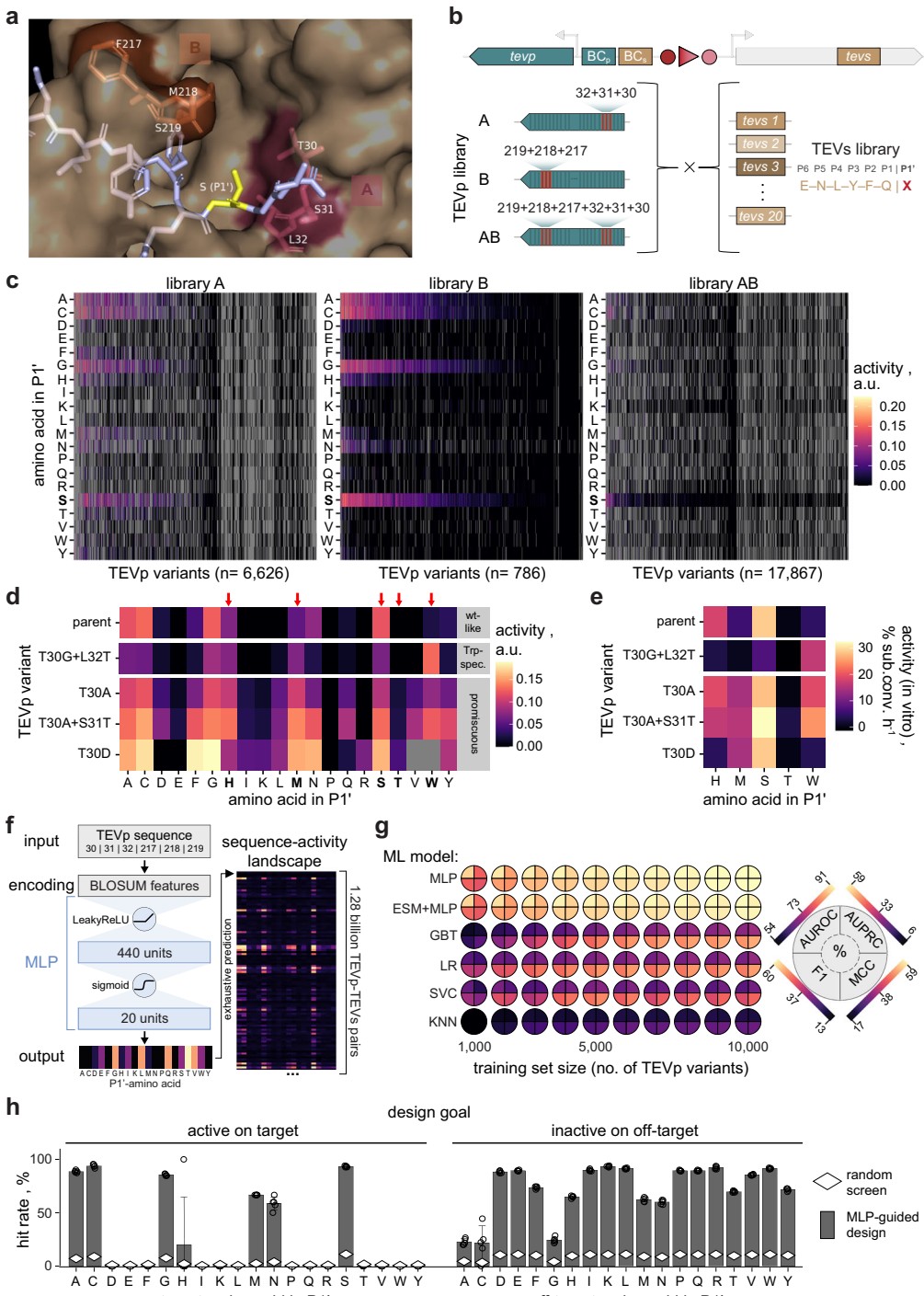

**Fig. 4 | Multi-site mutagenesis and ML-guided specificity engineering.**
**a** Randomized residues (cluster A: 30-32, cluster B: 217-219, stick model) are close to P1′ (yellow) in TEVs (stick model). Structure: PDB 1LVB[55]. **b** TEVp libraries randomizing either (A, B) or both clusters concomitantly (AB) were tested against all P1′-variants of TEVs (X = any amino acid). **c** P1′-specificity profiles for the libraries in (**b**). Variants are ranked according to mean activity on all tested substrates from high (left) to low (right). The canonical substrate ENLYFQS is highlighted (bold). Gray boxes indicate missing data. **d** Variants with distinct specificity were re-tested with the DNA recorder to obtain full P1′-profiles. A variant with a strong preference for W (Trp-spec.) and three with relaxed specificity (promiscuous) are compared to parent TEVp I. Characteristic TEVs variants (arrows, bold) were chosen for in vitro assays. Gray boxes indicate missing data. **e** In vitro activity of variants from (**d**) on characteristic P1′-variants. **f** MLP architecture. A multi-output classifier was used to model activity on 20 P1′-variants simultaneously using data from library AB (BLOSUM: Blocks Substitution Matrix). The architecture was optimized via systematic

hyperparameter search and renders exhaustive prediction of large sequence-activity maps feasible. A random subset of predicted specificity profiles from cluster AB is shown. **g** Predictive performance of different ML models trained on varying dataset sizes from library AB. AUROC: Area Under the Receiver-Operator Characteristic. AUPRC: Area Under the Precision-Recall Curve. F1: harmonic F-score. MCC: Matthews Correlation Coefficient. MLP: multilayer perceptron. ESM: evolutionary scale modeling. GBT: gradient-boosted trees. LR: logistic regression. SVC: support vector classifier. KNN: k-nearest neighbor classifier. Color indicates mean value over $n = 5$ model replicates with different seeds. **h** Efficiency of random versus MLP-guided screens using a test set of 1000 library AB variants. Hit rates are the fraction of variants that are active (left) or inactive (right) on the TEVs variant bearing the indicated P1′ amino acid. Right panel: only candidates retaining activity on canonical TEVs are displayed. Bars represent the mean for $n = 5$ replicate models with different random seeds with SD (error bars). Source data are provided as a Source Data file.

that ML-guided training rounds (i.e., active learning) can drastically improve success rates in enzyme engineering[36]. For TEVp, we sought to explore if prior knowledge about epistasis could be used to devise a smart sampling strategy that maximizes outcome at given experimental limitations. As previously shown, residues in close spatial proximity tend to interact stronger and more non-linearly[60], for which we also found indications in TEVp (Fig. 5a and Supplementary Fig. S17). We further investigated epistasis in the learned six-residue sequence-activity landscape mapped by the MLP model trained on library AB. This indicated that the epistasis within the spatially confined clusters A and B is indeed stronger than between the two clusters, which are more distant to each other (Fig. 5b and Supplementary Fig. S18). Building on this premise, we explored if creating a differential of epistasis within the training data could be leveraged to enhance learning. We propose to allocate the experimental budget with a bias towards clusters A and B, resulting in training data that is more densely sampled where epistasis is more prominent (Fig. 5c). Such emphasis on non-linearity could adaptively provide more data in regions where activity is harder to estimate for a model. We simulated if distributing a fixed experimental budget (data on 1200 TEVp variants and 9500 TEVp-TEVs pairs) differently amongst libraries A, B and AB had an impact on the predictive performance of ML models resulting thereof (Fig. 5c). Following this rationale, we trained different ML models both on a random sample from library AB ($1000_{AB}$ variants, 6500 TEVp-TEVs pairs) and a composite sample from all three libraries ($300_A$:$300_B$:$400_{AB}$ variants, 6500 TEVp-TEVs pairs). Models were evaluated on the same held-out test set of AB samples, which again revealed superior performance for the MLP (Fig. 5d and Supplementary Fig. S19). More importantly, comparing the sampling strategies consistently showed a strong performance advantage upon training with the composite sample, regardless of the selected ML approach, with absolute AUPRC improvements between 7% and 25% (average: 16.3%). This shows that training set design inspired by a priori epistatic considerations can indeed lead to significant performance increases in resulting ML models, which we could corroborate also for a different pair of structurally selected clusters of TEVp residues (Supplementary Fig. S20). We coined the corresponding epistasis-inspired sampling strategy ML-guided Directed Evolution with Epistatic Priors (MLDEEP), and performed an in-depth analysis of different MLDEEP compositions for the three libraries (Fig. 5e and Supplementary Fig. S21). While the optimal composition differed between models, the results show that high allocation of resources to library AB consistently leads to strong underperformance and that compositions with at least half of the samples allocated to the intra-cluster libraries are favorable. These coherent empirical findings over multiple models and datasets strongly speak for the use of epistatic priors in training set design. Best performances arise from the MLPs (with and without ESM) trained on approximately 80% intra-cluster and 20% inter-cluster data. Since this is dependent on the selected clusters and the protein of interest, we suggest the use of equivalent ratios between all libraries as a starting point for MLDEEP. These may be iteratively adapted for subsequent MLDEEP rounds based on obtained epistatic information in an active-learning manner.

## Discussion

In this study, we present a novel approach that leverages DNA recording to obtain sequence-activity data on proteases at very high throughput. The developed plasmid architecture enables parallel testing of each candidate on large numbers of potential on- and off-targets in parallel. As avoidance of off-target activity is of imperative importance[2], this addresses a major limitation of previously available tools for HTP protease engineering. We tested 29,716 TEVp variants on up to 134 substrates, each obtaining kinetic activity data on a total of approximately 600,000 TEVp-TEVs pairs. This represents the most extensive sequence-activity mapping effort for proteases reported to date. The presented

experimental workflow is fully parallelized and entirely PCR-free, which avoids biases due to sequential analysis of clones and amplification as previously shown[49]. Furthermore, elaborate experimental phenotyping procedures such as cell sorting or continuous cultivation are avoided, lending our approach a low accessibility barrier.

The obtained sequence-activity maps, in particular the 2D mutational scan combining all single-site mutants of both TEVp and TEVs, constitute a valuable resource suggesting potential starting points for specificity engineering. As our data suggest, promising avenues to that end include increasing P5-specificity, modulating specificity for P3 and P1' or relaxing substrate specificity by introduction of P1-promiscuity. While residue-wise randomization provides helpful indications on an enzyme's malleability for engineering, we emphasize that genuine re-purposing of protease specificity demands going beyond such approaches by applying multi-site mutagenesis. To this end, our data indicated strong epistatic interaction, which prompted us to simultaneously randomize six key TEVp residues in spatial proximity to P1' in TEVs and test the resulting library against all P1'-variants. The DNA recorder delivered sequence-activity data on over 264,000 TEVp-TEVs pairs, which we leveraged to exhaustively map the entire underlying sequence-activity landscape of $20^7$ TEVp-TEVs combinations by machine learning (ML). The comprehensive landscape can be efficiently searched for TEVp candidates with desired on- and off-target activity towards P1', leading to strongly increased hit rates compared to conventional random searches in the sequence space (Fig. 4h). Notably, neither the experimental nor the ML methodology alone would have led to some of the most critical findings. As Fig. 4g suggests, the large-scale data that our DNA recorder allows access to is indeed needed to obtain performant predictors. On the other hand, the drastically increased hit rates when comparing random with model-guided searches underline the potential of model-guided engineering. This highlights the potential of combining HTP experimentation and modeling by ML to efficiently search for improved variants in extremely large sequence spaces.

We further strengthened this symbiosis between DNA recorder and ML by proposing a novel, epistasis-aware strategy for training set design. The method, MLDEEP, relies on a priori available structural considerations to bias library design in favor of regions in the sequence space with a high likelihood of strong epistasis. MLDEEP leads to strong improvements of model accuracy compared to random sampling while being agnostic to the applied ML approach. Notably, this was achieved with libraries tested in independent experiments, indicating that data from different sites of a protein can be combined to obtain performant predictors by adding small combinatorial datasets that connect all sites. This could be capitalized on for collaborative efforts across different laboratories to jointly elucidate sequence-activity landscapes at a large scale. While showcased for P1'-specificity herein, the workflow may be used to tackle other positions in TEVs and other protease properties, such as stability or solubility. It could be augmented with unsupervised models trained on information about protein sequence, structure and folding to enhance performance and likelihood of success as previously shown[61–63]. Lastly, we expect that the epistasis-aware sampling procedure MLDEEP is applicable to most other proteins and thus of high value for the engineering and directed evolution of synthetic biological systems in more general terms.

## Methods
### Chemicals and reagents
Unless stated otherwise, chemicals were obtained from Sigma-Aldrich (St. Louis, Missouri, USA). Enzymes and enzymatic kits were obtained from New England Biolabs (Ipswich, USA). Oligonucleotides longer than 100 nt or with degeneracy were obtained PAGE-purified from Micro-synth (Balgach, Switzerland). All other oligonucleotides were obtained in desalted form from Sigma-Aldrich. Synthetic gene fragments were syn-thetized by Integrated DNA Technologies (Leuven, Belgium) or Twist Biosciences (San Francisco, USA). Synthetic oligo pools were obtained

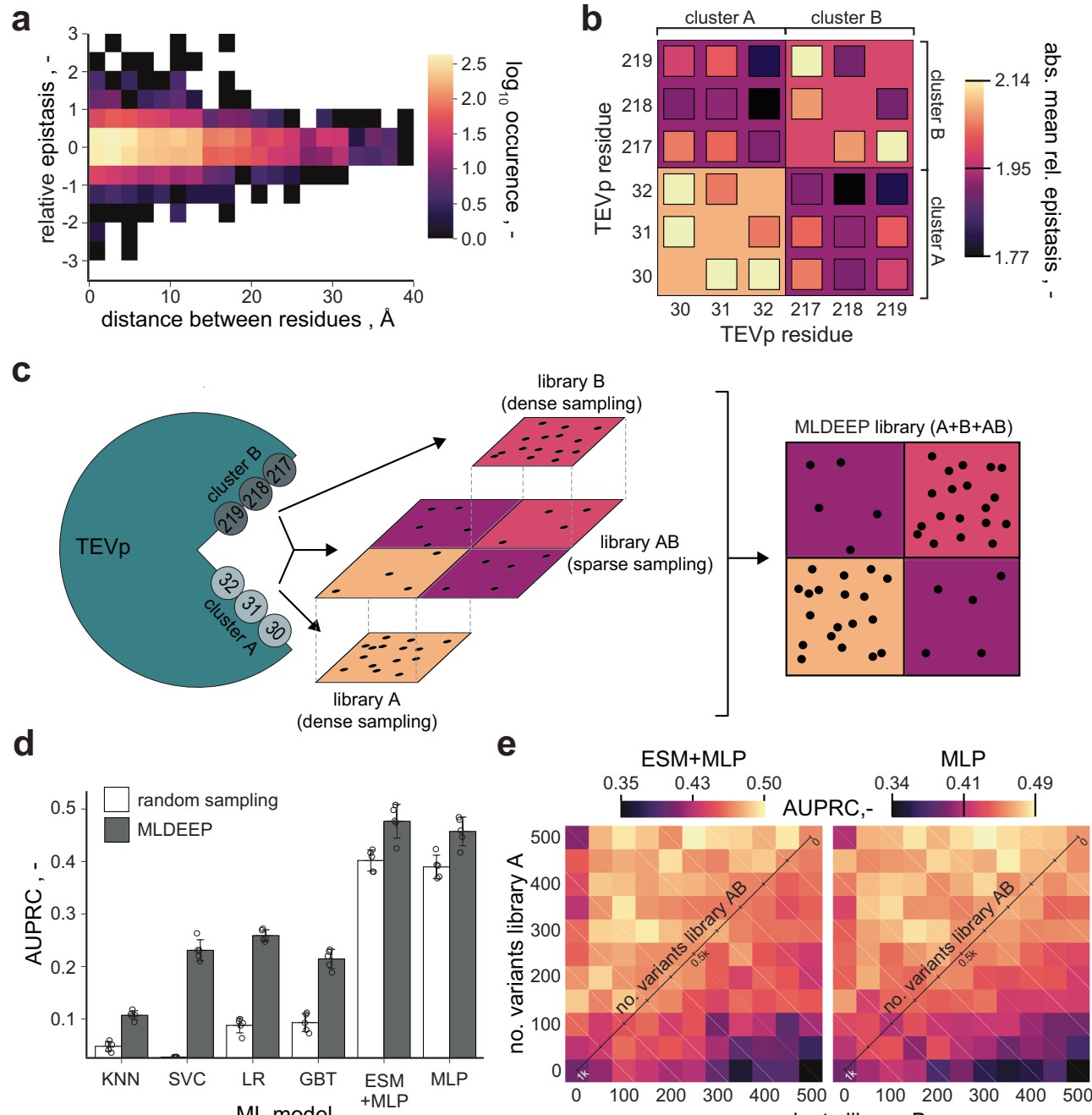

**Fig. 5 | Machine learning-guided directed evolution with epistatic priors (MLDEEP). a** Epistasis between TEVp residue pairs over their spatial distance (reference structure: 1LVB). Multiplicative epistasis was determined using the relative model described by Olson et al. 2014[60] and is displayed in bins of 0.5 and 2 Å for epistasis and distance, respectively. **b** Epistasis in the sequence-activity landscape learned by the MLP model trained on all available data of library AB. Epistasis between residues is calculated as in (**a**) and displayed as the absolute of the mean across residue (small squares) and cluster pairs (large squares). **c** Schematic representation of the MLDEEP rationale. Intra-cluster sampling (libraries A and B) is performed more densely than inter-cluster sampling (library AB) to leverage a differential of epistasis within and between the two clusters. Consequently, sampling is biased in favor of non-linear interactions in the sequence-activity landscape with the goal to increase learning efficiency at a given experimental budget. **d** Comparison of the performance (AUPRC) of

models resulting from training data obtained by random versus epistasis-aware (=MLDEEP) sampling. For random sampling, models were trained with data on 1000 TEVp variants from library AB. For MLDEEP, models were trained with data on 300 variants each from library A and B, as well as 400 variants from library AB. Bars indicate mean AUPRC of $n = 5$ replicate models initiated with different random seeds, with SD (error bars). **e** Analysis of different MLDEEP library compositions. A total experimental budget of 1000 TEVp variants was distributed in different ratios amongst libraries A, B and AB. The number of samples from library AB is indicated on the diagonal axis, i.e., the lower left and upper right squares correspond to 1000 or no samples from library AB, respectively. Performance for ESM + MLP and MLP is displayed as the mean AUPRC of $n = 5$ replicate models initiated with different random seeds. Source data are provided as a Source Data file.

from Twist Bioscience. Duplex DNA adapters were obtained from Integrated DNA Technologies. Plasmid DNA was prepared using the ZR Plasmid Miniprep kit from Zymo Research (Irvine, USA; cat. no. D4054) or, for NGS sample preparation, with the QIAprep Spin Miniprep kit (Qiagen, Hilden, Germany; cat. no. 27104). MetaPhor agarose was purchased from Lonza (Basel, Switzerland; cat. no. 50181). For clean-up of DNA from PCRs, restriction digests or ligations, the DNA clean & concentrator kit-5 from Zymo Research (cat. no. D4004) was used. DNA was extracted from agarose gels via the Zymoclean Gel DNA Recovery kit from Zymo Research (cat. no. D4002).

### Strains
Cloning, library generation and DNA recorder experiments were carried out with *E. coli* TOP10(DE3) Δ*rhaA* (F⁻ *mcrA Δ(mrr-hsdRMS-mcrBC) φ80lacZΔM15 ΔlacX74 nupG recA1 araD139 Δ(ara-leu)7697 galE15 galK16 rpsL(Str^R) endA1 λ⁻ ΔrhaA*), a derivative of TOP10(DE3) incapable of utilizing rhamnose[49]. For the production of cell free extracts, BL21(DE3) (F⁻ *ompT hsdSB(rB−, mB−) gal dcm* (DE3)) (Invitrogen; cat. no. C601003) was used.

### Cultivation of *E. coli*
Cells were routinely cultivated in lysogeny broth (LB) supplemented with 50 mg L⁻¹ kanamycin (cat. no. K1876) and 50 mg L⁻¹ streptomycin (cat. no. S9137). For strains containing DNA recorder plasmids, media additionally contained 10 g L⁻¹ glucose (cat. no. FLUH99C80E9C) to inhibit the rhamnose-inducible promoter controlling recombinase expression.

### Cloning procedures
Preparative PCR was performed using Q5 DNA polymerase (NEB; cat. no. M0491S) following the manufacturer's instructions. For library cloning, the cycle number was reduced to 15. Cloning was performed using conventional restriction-ligation cloning or the Gibson Assembly Master Mix (NEB; cat. no. E2611L) following the manufacturers' instructions. For libraries, the approximate number of transformants was determined by counting colonies on the respective plates using the image-based particle count feature of ImageJ (version 1.53a).

### Plasmids
Plasmids used in this study are listed in Supplementary Table S5. Plasmid pProtRec (Supplementary Fig. S22 and Supplementary Note 1) is based on pSEVA291 (kan^R, pBR322 ori)[64] and derived from parts of pASPIre3 (Addgene #154844)[49] and pASPIre4 (Addgene #196656)[50]. It embodies the optimized architecture for the DNA recorder for proteolytic activity used in this study and was used as a starting point for library generation and for the re-cloning of hits from libraries. It contains the Bxb1-sfGFP (superfolder green fluorescent protein) fusion used by Höllerer et al.[49] controlled by the weak RBS ACTCTGGATG-TAATGTG with an improved proteolytic degradation tag *SsrA^NYNY*[65] and a TEVs sequence with flexible adjacent linkers between sfGFP and the degradation tag. The two adjacent linkers were found to be superior to alternatives with only one or no linker (compare linker optimization in Supplementary Fig. S1). A map of the different linker constructs can be found in Supplementary Fig. S23. Further, the CDS of TEVp I controlled by a constitutive promoter J23107 is included in pProtRec. Sequences of several derivatives of pProtRec used as controls or to facilitate library generation are provided in Supplementary Notes 2–9. For in vitro experiments with cell-free extracts, individual TEVp variants were sub-cloned into plasmid pEXPrfp.

### Library generation
Generally, combinatorial TEVp-TEVs libraries were generated from half-libraries for TEVp and TEVs, respectively, which were joined restriction-ligation cloning. The TEVs P1' half-library was generated from 19 individually cloned P1'-variants (plus canonical ENLYFQS) with specific 6-nt barcodes[66]. The primers used for PCR-based generation of

these variants are listed in Supplementary Table S6. Individual variants were pooled in equimolar amounts, and the pool was joined with different TEVp libraries thereafter. The TEVs single-site variant half-library was generated from a synthetic oligo pool containing codons for all 19 non-wildtype amino acids for each TEVs position in equimolar ratios and an additional stop codon in position P1' (ENLYFQ*) as a positive control (max. signal). To facilitate cloning of the half-library, an *Sph*I site was introduced directly downstream of *tevs* in the recorder plasmid, which did not affect the DNA recorder output to a significant degree (Supplementary Fig. S24). In brief, the oligo pool was used as a forward primer in a PCR together with a reverse primer introducing a N₁₀-mer barcode (Supplementary Table S7). The PCR product was gel-purified and ligated into the pProtRec plasmid architecture. After transformation, approximately 1800 colonies were pooled as a glycerol stock before joining with TEVp half-libraries. The TEVp single-site variant half-library was based on parent variant TEVp 0, with each residue fully randomized to all 19 non-parent amino acids. For residues 213–234, a stop codon was additionally included to study the effects of truncation. The parent's nucleotide sequence is provided in Supplementary Note 10. The library was obtained in positional pools, which were further pooled into ten distinct bins (nine 24-residue pools and one 18-residue pools). The latter were individually PCR-amplified, adding N₁₅-mer barcodes (see Supplementary Table S8 for primers). The PCR product was purified and ligated into the pProtRec plasmid architecture using up to a 20-fold excess of insert to increase efficiency. Ligations were transformed, and 2600–4500 colonies were scraped from plates for each bin. Bins were combined by pooling according to their OD₆₀₀ to obtain the full single-site TEVp variant half-library. TEVp half-libraries A, B, C, AB and AC were generated by overlap extension PCR using pProtRec as template and primers introducing degenerate codons NNK or NNS at desired positions (Supplementary Table S9). The primer combinations to generate the individual DNA fragments are detailed in Supplementary Table S10, and fragments were joined using overlap extension PCR as outlined in Supplementary Table S11. The final PCR products introduced into the pProtRec architecture through restriction-ligation cloning before transformation. For TEVp half-library A, B and AB, ~ 300,000, 1100 and 10,000 colonies were scraped from plates and pooled, respectively.

Combinatorial TEVp-TEVs libraries were obtain by joining the respective TEVp and TEVs half-libraries through conventional restriction-ligation cloning. Transformed libraries were stored as glycerol stocks adjusted to an OD₆₀₀ of approximately 5, and subsequently used to directly to inoculate growth media for DNA recorder experiments.

### Fluorescence quantification of cellular Bxb1-sfGFP levels
Individual cultures were prepared by inoculating 2.5 mL LB in shake flasks from monoclonal overnight cultures to an OD₆₀₀ of 0.05. Cultures were incubated shaking (37 °C, 200 r.p.m.), and at an OD₆₀₀ of approximately 0.5, Bxb1-sfGFP expression was induced by adding 0.2% (w/v) ʟ-rhamnose (cat. no. W373011). After, 200 μL of induced cultures were transferred in triplicates into a sterile 96-well plate (flat bottom Nunclon Delta Surface, Thermo Fisher Scientific, Waltham, MA, USA; cat. no. 167574) and incubated in an Infinite® M1000 PRO plate reader (Tecan, Männedorf, Switzerland) at 34 °C (orbital shaking, amplitude of 1.5 mm). Fluorescence (excitation: 485 nm, emission: 510 nm) and OD₆₀₀ were measured every 15 min. To account for background fluorescence, uninduced cultures with 1% (w/v) glucose instead of rhamnose were included and used to blank the fluorescence signal. The OD₆₀₀ signal was blanked using plain medium.

### Workflow for DNA recorder experiments
For DNA recorder experiments, glycerol stocks of libraries were used to inoculate 600 mL of pre-warmed LB in 5 L baffled shake flasks to an initial OD₆₀₀ of 0.05. Cultures were incubated shaking (34 °C, 200 r.p.m.)

until an $OD_{600}$ of approximately 0.5 was reached, and induced by adding 0.2% (w/v) L-rhamnose. After, culture samples were taken in one-hour intervals, immediately centrifuged (10 min, 4000 x *g*, 4 °C), and pellets were rapidly frozen on dry ice. After, plasmid DNA was extracted and digested with *Spe*I and *Nco*I (37 °C, 4 h; NEB; cat. no. R3133S, R3193S). Target fragments for NGS containing barcodes for both TEVp and TEVs, as well as part of the Bxb1 substrate, were gel-purified and subsequently ligated to customized DNA duplex adapters (IDT) with sample-specific index combinations[49]. The final DNA fragments for Illumina NGS were again gel-purified using 2% MetaPhor agarose according to the manufacturer's instructions. Purity and concentration of the fragments was measured using capillary electrophoresis (Fragment Analyzer, Agilent), and fragments from the different samples were pooled equimolarly before subjecting the pools to Illumina NGS.

### Illumina sequencing and primary data processing

Illumina sequencing was conducted using NextSeq (150 cycles, paired-end) and NovaSeq 6000 (200 cycles, paired-end) platforms. Approximately 15–20% PhiX DNA was added to increase sequence diversity. Primary sequencing data were processed with Illumina RTA version V3.4.4 and bcl2fastq to obtain *.fastq files. Further data processing was performed using bash and R scripts (R version 4.2.2) executed on a Red Hat Enterprise Linux Server (release 7.9). Corresponding code was published earlier[49] and is available under: https://www.github.com/JeschekLab/uASPIre. This code was adjusted for this study. In brief, forward and reverse reads are paired, and the identity of the sample-specific indices as well as the state of the Bxb1 substrate (unflipped or flipped) are determined. After, reads are split into different libraries and sampling time points. TEVp and TEVs barcodes are extracted relying on adjacent constant sequences (20 nts each) located via fuzzy string matching, allowing up to three mismatches. Finally, reads are grouped by their combination of barcodes, and the fraction of flipped Bxb1 substrates is calculated for each TEVp-TEVs barcode pair to obtain time-resolved flipping profiles. To exclude low-quality data (i.e., sequences with low NGS coverage or read errors), only TEVp-TEVs pairs with at least 20 reads per sampling time point were considered. Further, we made use of cases in which TEVp-TEVs pairs were represented by two or more unique barcode combinations, which essentially represent biological replicates. In cases of three or more of such replicates, we compared the individual flipping profiles to identify and remove putative false positives. Specifically, we calculated the average flipping profile both in and excluding the top replicate. If removal led to a more than two-fold drop in the $AUC_{4h}$, the top replicate was considered an outlier and removed (approximately 0.3% of TEVp-TEVs pairs in the 2D-scan library). In cases of duplicates, TEVp-TEVs pairs were removed entirely if the top replicate had a greater than three-fold higher $AUC_{4h}$ than the bottom one. For more details, please refer to the annotated code provided with this submission.

### Barcode-to-variant assignment via long-read NGS

TEVp variant sequences of library A were assigned to their barcodes using Illumina sequencing. For this, the plasmid region containing codons 30–32 of TEVp and the TEVp barcode was PCR-amplified with primers listed in Supplementary Table S11. The PCR was conducted using 125 ng of library plasmid pool as template for 15 cycles, and amplicons of the correct size were gel-purified and subjected to Illumina NGS (150 cycles, paired end). Variant sequences of the other TEVp (libraries B, AB and single-site variants) and TEVs (single-site variants) libraries were assigned to the corresponding barcodes using long-read NGS (SMRT-seq, PacBio). For this, fragments spanning TEVp CDS and barcode or TEVs CDS and barcode were excised by restriction digest using *Asc*I (NEB; cat. no. R0558S) and *Nco*I or *Age*I (NEB; cat. no. R3552S) and *Not*I (NEB; cat. no. R3189S), respectively. Fragments were gel-purified from unstained agarose gels, and DNA concentrations were measured using capillary electrophoresis (Fragment Analyzer, Agilent). Individual samples were pooled based on

determined concentrations according to the expected library size and desired sequencing depth. Final NGS was performed on a Sequel II platform (SMRT-seq, PacBio) using the SMRTbell prep kit 3.0.

To assign TEVs sequences to the corresponding barcodes, a string-matching approach with constant sequences flanking the randomized parts was implemented. In brief, a bash script is first used to identify the positions of constant sequences, after which an R script to extract TEVs sequences and barcodes is applied. After, TEVs DNA sequences are translated into amino acid strings and unique barcode-sequence combinations are grouped and counted. To assign TEVp sequences with barcodes, an alignment strategy based on minimap2 and alignparse was applied inspired by previous work[67]. Briefly, a Pthon script is used to align SMRT-seq reads to a reference sequence, which consists of the CDS of parent variant TEVp 0 linked to constant sequence regions as well as a degenerate $N_{15}$ sequence as a placeholder for the TEVp barcode. Subsequently, TEVp mutations and barcodes are extracted, and unique barcode-sequence combinations are grouped and counted. For more details, please refer to the annotated code provided with this submission.

### Selection of variants with new P5-specificity

Variants from the 2D-scan library with new P5-specificity were chosen considering only candidates appearing with at least ten different P5 variants. From those, we selected candidates with a mean activity on all tested P5 variants amounting to less than 40% of the activity of the parent TEVp 0. Finally, we selected from the resulting subset variants that with an activity greater 0.03 on at least one and at most three of the tested P5 variants.

### Machine learning

**Data preparation.** For ML analyses, DNA recorder data was rearranged into two tensors $X \in \mathbb{R}^{NxPxD}$ and $Y \in \mathbb{R}^{NxS}$, representing the model input (TEVp variants' sequences) and the prediction target (activity profiles on TEVs variants), respectively, where $N$ is the number of TEVp variants (= 17,863), $P$ the number of randomized TEVp positions (= 6), $D$ the number of embedding dimensions (= 20), and $S$ the number of TEVs substrate variants (= 20). TEVp sequences were represented by the six randomized TEVp residues only (input tensor $X$). Stop codon mutations (result of NNK/NSS degeneration) were excluded. The target tensor $Y$ (label) was constructed by concatenating the $AUC_{4h}$ values for each TEVs variant while indicating missing values, which were later on masked in the model (see below). Tensor $Y$ was further accompanied by a tensor $R \in \mathbb{R}^{NxS}$ holding the minimum per-timepoint NGS read count for each TEVp-TEVs pair as an additional metric for data quality.

Target labels $Y$ were further discretized using a hysteretic threshold derived from negative controls (compare Supplementary Figs. S5 and S7). Specifically, we computed *p*-values for each variant based on a normal distribution fitted on the negative controls and subsequently used them to categorize TEVp variants as inactive (= 0) or active (= 1) in a threshold-based fashion. To avoid edge effects, we defined a more ($p_1 = 0.01$) and a less stringent ($p_2 = 0.1$) threshold. If for a given TEVp variant the activity on any TEVs variant surpassed the more stringent threshold $p_1$, all activity values were binarized using the less stringent threshold $p_2$. This strategy was employed to emphasize weak off-target activities and prevent their misclassification as a result of noisy data.

**Models.** MLPs were developed in a hyperparameter optimization. The base architecture of neural networks consisted of $l$ fully connected hidden layers with $h$ units followed by a LeakyReLU[68] non-linear activation function and a dropout layer[69] with dropout probability $p$. The input TEVp sequence was encoded with an embedding $\omega$, which was either a one-hot or one of several BLOSUM encodings (of dimension 20). The input dimension of the MLP corresponds to the length of the concatenated embeddings of the input sequence. The final layer is a

fully connected layer with 20 units, one for each substrate (i.e., the model predicts the full substrate profile of TEVp). The MLP was trained under a focal loss objective $\mathcal{L}_{Focal}(p_t) = -\alpha(1-p_t)^\gamma \log(p_t)$[70] with two parameters $\alpha$ and $\gamma$ that control the importance of the positive class, and the weight of samples close to the decision boundary of the model, respectively. Both losses were sample-weighted with a factor $w = (r/c)^\rho$ where $r$ is the read count of the activity measurement, and $c, \rho$ are hyperparameters. This read count-based weighting introduces the notion of data quality and measurement confidence into model training, where a value of $\rho = 0$ is equivalent to no weighting. In addition, the minimum read count threshold $\theta$ for quality control on the training data was part of the hyperparameter optimization. Training was terminated using early stopping with a patience of 300 steps. As both model and data have small memory footprints, models were trained using batch gradient descent, i.e., the mini-batch size was set to the size of the dataset. All models were implemented in PyTorch[71] (version 2.4.0 + cu118) using the Adam optimizer[72] with learning rate $\eta$ and otherwise default parameters. All models were run on 10GB MIG instances of H100 NVIDIA GPUs. Hyperparameter optimization was conducted with Optuna[73] (version 3.6.1) over 100 trials using 10 concurrent workers, the hyperparameter ranges and best values are detailed in Supplementary Table S13. The optimization objective was set to maximize the average AUPRC over three replicates using a tree-structured Parzen estimator with default parameters. Trials were pruned after a maximum duration of one hour.

The ESM model (variant esm2_t33_650M_UR50D) was taken from the official repository at https://github.com/facebookresearch/esm, using pretrained weights. The model was trained by fine-tuning it with a prediction head of the same architecture as the MLP attached to the last layer, and otherwise frozen weights. We reduced the dimensionality of last-layer ESM embeddings to 120 by Principal Component Analysis (fitted on the training set) before feeding them to the prediction head, as suggested for variant effect prediction by the authors. Per-token embeddings were concatenated. Hyperparameter optimization was conducted analogous to the MLP, the final configuration is detailed in Supplementary Table S14. The remaining models (GBT, LR, SVC, KNN) were implemented using scikit-learn[74] (version 1.5.1). To simplify the multi-output setting of our prediction problem for these models, the prediction was phrased as a single-output classification by concatenating embeddings of TEVp with a one-hot embedding of TEVs as input to the model (i.e., the model predicts the activity of each TEVp-TEVs pair). Hyperparameter ranges and best values from the optimization can be found in Supplementary Tables S15–S18.

**Evaluation.** For training, validation, optimization, and testing, the data was split into multiple partitions on a TEVp-variant level. To properly assess model performance, we required that the split partitions used in evaluation had activity measurements on all substrates and a certain minimal quality in terms of NGS reads. We, therefore, randomly sampled test, validation, and stop data from the subset of completely measured TEVp variants with at least 100 reads per substrate. Out of the 1625 TEVp variants fulfilling these criteria, the sets were composed out of 1000, 300 and 300 variants, respectively. Validation and stop sets were sampled with five different random seeds to yield a better estimate of performance during optimization and validation, and were merged into the training data after optimization or evaluation, respectively. Classification models were evaluated using F1, MCC, AUPRC, and AUROC. AUPRC is intended for imbalanced classification problems and is the primary performance metric used herein, with a positive class prevalence of 2.75% in the test set. Note that there were no common TEVp variants in training and test sets, and that >80% of test set variants had a Levenshtein distance of at least two (out of six) to the closest variant in the training set.

**Variant design.** To design novel protease variants with a desired activity pattern, we exhaustively predicted the P1′-activity profiles of all theoretically possible variants in the combinatorial sequence space ($20^6$ TEVp variants) and then scored them with respect to the desired target profile. Scores were computed using an inner product of the output probability vector and the target profile vector, consisting of values of 1, 0 and -1 for activity, irrelevance and inactivity for the different TEVs variants, respectively. To evaluate the model's practical utility for variant design, we first determined the prevalence of variants with desired target profiles in the test set. Analogously, we determined the fraction of variants that actually showed the desired target profiles amongst variants predicted to have it according to the model. This procedure emulates, based on the experimental test set, how efficient a model-guided search would be in comparison to a random experimental search in the sequence space.

**Epistasis analyses.** To analyze the relation of epistasis and spatial distance in TEVp we relied on additional double mutations, which accidentally occurred in the 2D-scan library, likely due to PCR-mediated recombination between single mutants. We computed the epistatic effect $\varepsilon = \ln(W_{ij}) - \ln(W_i) - \ln(W_j)$, where $W_{ij}$ is the activity of the double mutant carrying mutations $i$ and $j$ normalized to the parent's activity, and $W_i$ and $W_j$ are the normalized activities of the two single mutants carrying mutations $i$ and $j$, respectively. This follows the epistasis model of Olson et al.[60] and Khan et al.[75], who conducted similar studies and inspired our approach. The epistasis values were plotted over Euclidean distances between the residues' center of mass as extracted from the reference structure PDB 1LVB. To analyze the epistasis in the learned sequence-activity landscape, we predicted the activity of all theoretically possible single and double mutants with one instance of the MLP model that was trained on the entire AB library. Epistasis was then computed analogously as described above.

**MLDEEP.** To construct different composite libraries, we randomly sampled defined numbers of TEVp variants from libraries A, B and AB to a sum of 1000 variants in total, using a fixed read threshold $\theta = 100$ and subsequently down-sampling TEVp-TEVs pairs to 6500 by random masking. The latter allows for a fair comparison at given fluctuations of TEVs variants tested per TEVp variant, ensuring that all models are trained on the same amount and quality of data. Evaluation was conducted on a held-out test set of 1000 TEVp variants with full substrate coverage from the library AB, which was identical across all models and MLDEEP partitions.

### In vitro assay for TEVp activity

In vitro assays and plasmids to test selected TEVp variants are based on previous work[53] and relied on TEV substrates exhibiting Förster resonance energy transfer (FRET) expressed from plasmid pFRET. TEVp variants were fused to mCherry (for normalization to protein concentration) expressed from plasmid pEXPrfp. Both FRET substrates and TEVp variants were produced and used as cell-free extracts (CFXs). The FRET substrates comprised a translational fusion of a cyan (CFP[76]) and a yellow fluorescent protein (YFP[77]) linked by a peptide containing TEVs. To produce the corresponding CFXs, monoclonal LB pre-cultures of E. coli BL21(DE3) with the respective pFRET plasmid variant were diluted 1:100 into 5 mL of autoinduction medium ZYM-5052[78] and incubated overnight at 30 °C, shaking. Cells were collected (4000 rcf, 20 min, 4 °C) and frozen at −20 °C before thawing and resuspension in 10 mL (g of wet cell weight)$^{-1}$ of TE buffer (cat. no. 8890-OP) with 1 mM DL-dithiotreitol (DTT; cat. no. D0632). Cells were subsequently lysed by incubation with lysozyme (2 g L$^{-1}$, 45 min, room temperature; cat. no. 10837059001) followed by a subsequent freeze-thaw cycle. After, 1.4 mM MgCl$_2$ (cat. no. M8266) and 40 μg L$^{-1}$ DNase I (cat. no. 11284932001) were added and samples were incubated for 20 min to

reduce viscosity. Finally, 1.4 mM EDTA (cat. no. E6758) was added and the lysate was cleared by centrifugation (4000 x g, 25 °C, 20 min). The supernatant was heat-treated in a water bath (60 °C, 15 min) and precipitated proteins were removed by centrifugation (4000 x g, 4 °C, 20 min). The resulting cleared CFX containing the CFP-YFP quencher pair was stored at 4 °C until further use. TEVp variants were produced analogously using plasmid pEXPrfp at a ten-fold higher volume compared to the production of FRET substrates. Here, the lysate was cleared by centrifugation at 4000 x g and 25 °C for 10 min. Concentrations of FRET substrates and TEVp variants in both CFX types were determined through calibration curves using purified mCherry and FRET substrate in TE buffer supplemented with 1 mM DTT, respectively.

The final in vitro activity assay is based on a sharp decrease in FRET efficiency upon TEVp-mediated hydrolysis of the TEVs linker between CFP and YFP. YFP fluorescence observed upon direct excitation of the YFP chromophore is indicative of the total substrate amount, while fluorescence upon excitation of CFP enables quantification of cleavage. The ratio of CFP-to-YFP fluorescence provides a robust, internally normalized metric and were calculated relative to a negative control lacking TEVp activity (i.e., $\Delta$(CFP/YFP)). Moreover, positive controls employing proteinase K (cat. no. P2308) from *Tritirachium album* (0.17 µM, in TE), resulting in full cleavage of TEVs without affecting CFP or YFP fluorescence, were included. Thus, the fraction of cleaved substrate relative to the total substrate amount can be determined. Furthermore, the mCherry signal serves as an indicator of TEVp concentration that can be used to approximate specific activities in the CFXs. Fluorescence was measured in black microtiter plates (Greiner 96 Flat Bottom Black Polystyrene, Greiner, Kremsmünster, Austria; cat. no. 655090) using a Tecan Infinite M1000 PRO plate reader (Tecan, Männedorf, Switzerland). The specific wavelengths for excitation and emission (Ex/Em) were 590 nm/610 nm for mCherry, 440 nm/490 nm for CFP, and 520 nm/530 nm for YFP (bandwidth: 5 nm). The final assay conditions were 0.6 µM TEVp variant, 0.9 µM FRET substrate, and 1 mM DTT at 37 °C. A linear regression analysis was conducted using data points within the first two hours after reaction start, using the slope as metric for the initial reaction rate.

### Reporting summary

Further information on research design is available in the Nature Portfolio Reporting Summary linked to this article.

## Data availability

The main data generated in this study are provided in the Supplementary Information and Source Data file. Raw NGS data generated in this study have been deposited in the NCBI SRA database under accession code SAMN48276352. Processed data are available at https://github.com/JeschekLab/ProtRec/ and https://github.com/BorgwardtLab/MLDEEP under CC-BY-4.0. Source data are provided in this paper.

## Code availability

Code used to develop perform the analyses and generate models and results in this study is publicly available and has been deposited at https://github.com/JeschekLab/ProtRec/[79] and https://github.com/BorgwardtLab/MLDEEP[80] under MIT. The specific version of the code associated with this publication is archived at Zenodo and is accessible via https://doi.org/10.5281/zenodo.15346003 and https://doi.org/10.5281/zenodo.15344074.

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

## Acknowledgements

We kindly thank Dr. Luzius Pestalozzi for providing plasmids encoding the different TEVp variants, as well as purified TEVp and FRET substrate. Further, we want to thank Tianyu Xu for assisting in the production of cell-free extracts. Finally, we thank all members of the Genomics Facility Basel, especially Dr. Christian Beisel and Ina Nissen-Naidanow, for support with NGS experiments. This work was kindly supported by the Swiss National Science Foundation (SNF) (project: Xenoimport; grant no. 179521, S.P.), the National Center of Competence in Research (NCCR) in Molecular Systems Engineering (MSE, S.P.), and the European Commission (project: NEWmRNA; grant no. 965135, S.P. & ERC project BiosenSAI: grant no. 101117399, MJ).

## Author contributions

M.J. and L.H. conceived the project. M.J. coordinated the study. K.B. conceived and supervised machine learning analyses. M.J. and S.P. supervised experimental work. L.H. performed experiments. L.H., S.H., and T.K. analyzed data. S.H. performed NGS raw data processing. S.H. and L.H. developed the computational pipeline for processing of NGS data to generate variant-barcode lookup tables. T.K. conceived, developed, and analyzed machine learning models and the MLDEEP method. L.H., M.J., and T.K. wrote the manuscript with input from all authors.

## Funding

## Competing interests

The authors declare the following competing interests: K.B. is co-founder and scientific advisory board member of Computomics GmbH, Tübingen. T.K. is co-founder and managing director of DropSort UG & Co. KG, Kronberg. The other authors declare no competing interests.
