## [Transparent Peer Review file · Nature Communications]

Data-driven Protease Engineering by DNA-Recording and Epistasis-aware Machine Learning

Corresponding Author: Professor Markus Jeschek

Version 0:

Reviewer comments:

Reviewer #1

(Remarks to the Author)
Comment to authors:

My impressions upon review of manuscript NCOMMS-25-02123-T, entitled "Data-driven Protease Engineering by DNA-Recording and Epistasis-aware Machine Learning" by Huber et. al., is that they have obtained an enormous amount of data for TEVp and TEVs mutations and their specific matched pairs using NGS and ML analysis. This was done efficiently and presented elegantly. I find this an excellent and ambitious piece of work. Readers' should be highly interest in this manuscript which resulted in a library of TEVp-TEVs pairs for drug discovery and therapeutical usage, as well as applying their technique for discovery of other protease-substrate pairs. Overall, this is splendid work considering the amount of data and size of their effort. I would recommend accepting it with minor revisions.

Abstract:
No issues

Introduction:
No issues

Result:
Line 85: typo, need s for "ubstrate"
Line 148: Please add definition of "possible" in this article. This would be useful as this word is frequently used in the results section, especially in 2D mutational scan of proteases-substrate pairs.
Line 150: "relaxed P1'-specificity", like above, a brief description of "relaxed" would be helpful for readers.
Line 219: there is an extra "that" before "potentially"
These authors presented discovery of an enormous number of TEVp - TEVs (P1') using ML. How was the result of ML evaluated (accuracy)? Was there any overlapping results with sequence-activity outcomes? Such data will encourage readers to apply ML methods.

Discussion:
Line 392: The total number of substrate variants is indicated as 134 in the results.
What is your hypothesis for the increase in TEVp activity when linker number increases? Would it be related to the protein configuration? Please add a line or two to address these questions.
One of the main points of this article was to discover how TEVp candidates cause off-target activity. There was no discussion about this. It would be advisable to address this based on the results from NGS.
What are pros and cons of each methodology based on NGS and ML? or is a combination of the NGS and ML necessary?

Figures and tables:
No issues

References:
"As a guideline, articles are allowed up to 70 references", the total number of references is 81.

Supplement:
All good.

(Remarks on code availability)

The authors have indicated that: "All data and code required to reproduce the analyses and figures are provided with this submission and will be made available in public repositories upon publication."

Reviewer #2

(Remarks to the Author)

(Remarks on code availability)

Reviewer #3

(Remarks to the Author)

In this paper, the author presents a DNA recorder for deep specificity profiling of proteases and proposes epistasis-aware training set design as a generalizable strategy to streamline searches within enormous sequence spaces to improve model accuracy. I have several critical comments for this manuscript that need to be fixed.

1. Regarding "...high substrate specificity is crucial to avoid off-target cleavage and thus undesirable side effects", it is recommended to clarify the specific impact of "off-target cleavage" on drug application.
2. When introducing the "recombinase-based DNA recorder", it is recommended to provide a clearer description of the mechanism.
3. The concept of "epistasis-aware" is rather technical. It is recommended to briefly explain its principle or provide reference support.
4. In the introduction section, the author mentioned "To avoid emergence of proteolytic promiscuity, the use of counterselection substrates, mainly the protease's native substrate, has been suggested..." and cited 9 references, which gave a relatively general description. It is recommended that the author add some analysis content such as the specific advantages or disadvantages of the method when discussing this point.
5. For exact numbers such as "64 million protease sequences", it is recommended to state the source of the data or add references to increase credibility.
6. Please check for typos, e.g. "ubstrate" in the subsection "Recording proteolytic activity within DNA".
7. Please add references for "Interestingly, while mutation of substrate-interacting residues was expectedly prone to activity loss...".
8. The author mentioned, "possibly due to a widened substrate pocket upon W211 replacement." What is the basis for this conclusion? Please add references.
9. It is recommended to add relevant descriptions for the figures to help readers understand the figure. For example, what is the corresponding relationship between the color range changes and activity in heat maps 2b and 2c?
10. The author mentioned that "we framed the model as a multi-output classifier of binary activities on different substrates to account for class imbalance." Does it mean that 135 models were trained? When training GBT, LR, SVC, and KNN models, were the embeddings of TEVp and TEVs concatenated to train a single model?
11. The distribution of mutations in the dataset has an intuitive impact on the performance of the model. It is recommended that the authors add experiments to analyze the performance of the model from different mutation distribution perspectives.
12. How is the test set constructed? Can the generalization ability of the model be verified? It is recommended that the authors use independent test sets of data from other proteases to verify the generalization ability of the model.
13. What information does the "target activity profile" contain? Authors are advised to provide detailed information in the data preparation section.
14. The input dimension (6 or others) of MLP should be related to the length of the input sequence. It is recommended that the authors clearly state this in the "Models".
15. The embeddings obtained from ESM should be two-dimensional. How did the authors convert them into input for MLP?
16. As shown in Figure 5d, MLDEEP has a significant improvement over SVC, LR, and GBT, but the improvement over KNN, ESM+MLP, and MLP is slightly weaker. It is recommended that the authors analyse this point.

(Remarks on code availability)

Version 1:

Reviewer comments:

Reviewer #1

(Remarks to the Author)

The authors have satisfied all concerns on this reviewer's part. Additional modifications made to the main text are more reader friendly and reinforce a better understanding of the authors' statements. I think the article can be accepted as it is.

(Remarks on code availability)

Reviewer #2

(Remarks to the Author)

(Remarks on code availability)

Reviewer #3

(Remarks to the Author)

The authors addressed my comments

(Remarks on code availability)

Reply to Reviewers' comments

NCOMMS-25-02123-T

Data-driven Protease Engineering by DNA-Recording and Epistasis-aware Machine Learning

Point-by-point responses are provided in-line with the reviewers' comments and *highlighted in blue font*.

The indicated line and figure numbers refer to those in the new revised version of the manuscript.

REVIEWER COMMENTS

Reviewer #1 (Remarks to the Author):

Comment to authors:

My impressions upon review of manuscript NCOMMS-25-02123-T, entitled " Data-driven Protease Engineering by DNA-Recording and Epistasis-aware Machine Learning" by Huber et. al., is that they have obtained an enormous amount of data for TEVp and TEVs mutations and their specific matched pairs using NGS and ML analysis. This was done efficiently and presented elegantly. I find this an excellent and ambitious piece of work. Readers' should be highly interest in this manuscript which resulted in a library of TEVp-TEVs pairs for drug discovery and therapeutical usage, as well as applying their technique for discovery of other protease-substrate pairs. Overall, this is splendid work considering the amount of data and size of their effort. I would recommend accepting it with minor revisions.

The authors kindly thank Reviewer #1 for their very positive feedback on our study and their effort in reviewing our manuscript.

Abstract:

No issues

Introduction:

No issues

Result:

Line 85: typo, need s for "ubstrate"

Typo corrected.

Line 148: Please add definition of "possible" in this article. This would be useful as this word is frequently used in the results section, especially in 2D mutational scan of proteases-substrate pairs.

The term "possible" in this context was meant to refer to the theoretically possible variants/mutations. We agree with the reviewer that this may lead to confusion and have redacted all instances of the term in the main manuscript and SI.

Line 150: "relaxed P1'-specificity" , like above, a brief description of "relaxed" would be helpful for readers.

The term "relaxed [...] specificity" in this context refers to the acceptance of or activity on an increased number of amino acids for a given residue. We agree with the reviewer that this may lead to confusion and have added a definition upon first use in the main text of the manuscript (line 155).

Line 219: there is an extra "that" before "potentially"

Typo corrected.

These authors presented discovery of an enormous number of TEVp - TEVs (P1') using ML. How was the result of ML evaluated (accuracy)? Was there any overlapping results with sequence-activity outcomes? Such data will encourage readers to apply ML methods.

Because of the imbalance in the data (there are much more inactive variants than active), Accuracy as a metric would be misleading. We thus chose to evaluate models with AUPRC, AUROC, F1, and MCC, which are better suited for imbalanced problems. The ML models' performance with respect to these metrics is for instance shown in Figure 4g in relation to training set size. These metrics (e.g. AUPRC up to 0.59) indicate substantial overlap of predictions with experimental measurements. However, as these metrics are less intuitive than accuracy, we have added an additional exemplary plot comparing prediction with measurement directly (new panel e in Fig. S16). Here, one can see that the probability output predicted by the MLP aligns well with the measured activity profiles across the different tested P1' variants of TEVs. We note, however, that this is a rather anecdotal comparison and that for a systematic comparison of measurement and prediction the aforementioned metrics are more meaningful.

Discussion:

Line 392: The total number of substrate variants is indicated as 134 in the results.

The reviewer is correct, it should be 19x7 single-site mutants plus wildtype substrate → 134 substrates. We corrected this typo in the manuscript.

What is your hypothesis for the increase in TEVp activity when linker number increases? Would it be related to the protein configuration? Please add a line or two to address these questions.

We agree with the reviewer that this observation is indeed interesting. Our underlying hypothesis for the addition of linkers was that they could make the substrate more easily accessible for TEVp (compared to a situation where it is "buried" in a structured part of the protein. We had added a remark to that end in the previous version of the manuscript (lines 106-107: "...we optimized accessibility of TEVs adding flexible amino acid linkers flanking the protease substrate..."). Additionally, we now added an additional statement on this hypothesis to the corresponding display item (see caption of Fig. S1).

One of the main points of this article was to discover how TEVp candidates cause off-target activity. There was no discussion about this. It would be advisable to address this based on the results from NGS.

We agree with the reviewer that addressing off-target activity is a key feature of our approach, more specifically the fact that we can measure the activity on multiple substrates simultaneously already during the HTP steps. An excellent example to that end are the variants that were found in the NGS data to have an increased specificity for P5 in the substrate, for which the wildtype protease is virtually fully promiscuous (Fig. 3b). While this is arguably an anecdotal example without a concrete application in mind, it highlights the potential of the ability to test for multiple substrates simultaneously (note that these more specific variants with less "off-target" effects could not have been identified in a conventional screen). To highlight this important point more, we have added a statement to the respective section (lines 236-238) and kindly thank the reviewer for this insightful comment.

What are pros and cons of each methodology based on NGS and ML? or is a combination of the NGS and ML necessary?

We thank the reviewer for this insightful question and would like to point out that we are convinced that it is the latter, i.e. neither the experimental nor the ML methodology alone would have led to some of the most critical findings in our study. As can be appreciated from Fig. 4g, the experimentally obtained training data is needed to build good in silico models. A strong indication to that end is that an increasing training set size leads to higher predictive performance. In other words, the big data that our DNA recording techniques allows access to is needed to obtain performant predictors. On the other hand, several critical and non-trivial findings underline the importance and power of model-guided engineering in our study. An example to that end are the drastically increased hit rates when comparing random with model-guided searches (Fig. 4h). Moreover, the results displayed in Fig. 5 (training set design based on

epistatic priors) highlight how a mutual interplay of experimental design and machine learning can be leveraged to make the search for improved variants within enormous sequence spaces significantly more efficient (see e.g. Fig. 5d). To better highlight the importance of the combination between experimental and computational methodology, we added corresponding sentences to the discussion (lines 421-427) and thank the reviewer for this insightful question.

Figures and tables:

No issues

References:

"As a guideline, articles are allowed up to 70 references", the total number of references is 81.

For the main manuscript, there are a total of 64 references whereas the remaining ones are only relevant for Methods and SI. If possible, we would like to maintain the references to appropriately acknowledge prior art. However, if needed we'd be happy to reduce the number of references and would leave it to the editors' discretion to make a decision to that end.

Supplement:

All good.

Reviewer #1 (Remarks on code availability):

The authors have indicated that: "All data and code required to reproduce the analyses and figures are provided with this submission and will be made available in public repositories upon publication."

Indeed, data and code will be made publicly available through Github repositories upon publication.

Reviewer #2 (Remarks to the Author):

The authors kindly thank Reviewer #2 for their effort in reviewing our manuscript and wish them all the best for their future career.

Reviewer #3 (Remarks to the Author):

In this paper, the author presents a DNA recorder for deep specificity profiling of proteases and proposes epistasis-aware training set design as a generalizable strategy to streamline searches within enormous sequence spaces to improve model accuracy. I have several critical comments for this manuscript that need to be fixed.

The authors kindly thank Reviewer #3 for their feedback on our study and their effort in reviewing our manuscript. We have adapted the manuscript following the feedback of this reviewer as detailed in the point-by-point responses below.

1. Regarding "...high substrate specificity is crucial to avoid off-target cleavage and thus undesirable side effects", it is recommended to clarify the specific impact of "off-target cleavage" on drug application. *When using proteases in or as drugs, it is imperative that they only cleave the target but not any of the patient's other proteins. The latter might lead to side effects and should thus be avoided at all costs. We clarified this by adapting the indicated statement in the introduction (lines 38-39).*

2. When introducing the "recombinase-based DNA recorder", it is recommended to provide a clearer description of the mechanism.

The term "recombinase-based DNA recorder" is introduced at the end of the introduction with the sole purpose to list the achievements of the study on a high level. A detailed description on the concept and

mechanism of the recorder is provided directly thereafter at the beginning of the results section (lines 80-101). We believe these longer descriptions are very important for understanding and yet, as they are one of the main conceptual results of the paper, are better placed in the results section.

3. The concept of “epistasis-aware” is rather technical. It is recommended to briefly explain its principle or provide reference support.

We have coined this term to highlight that epistatic priors can be used to increase the informativeness of experimentally generated training data. To make this more understandable, we clarified this upon first use of the term in the introduction (line 76).

4. In the introduction section, the author mentioned “To avoid emergence of proteolytic promiscuity, the use of counterselection substrates, mainly the protease’s native substrate, has been suggested...” and cited 9 references, which gave a relatively general description. It is recommended that the author add some analysis content such as the specific advantages or disadvantages of the method when discussing this point.

The motivation behind this statement was to point to an apparent lack in the state of the art of HTP protease screening in terms of the ability to measure activity on many substrates in parallel. Using a different term here (promiscuity instead of off-target activity) may indeed lead to confusion, which is why we re-phrased this statement to be consistent with terminology (line 62). Crucially, our approach specifically addresses the aforementioned lack in the state of the art enabling parallel assessment on a large number of substrate variants in parallel. An excellent example to that end are the variants that were found in the NGS data to have an increased specificity for P5 in the substrate, for which the wildtype protease is virtually fully promiscuous (Fig. 3b). This highlights the potential of the ability to test for multiple substrates simultaneously (note that these more specific variants with less “off-target” effects could not have been identified in a conventional screen). To highlight this important point more, we have added a statement to the respective section (lines 236-238) and kindly thank the reviewer for this insightful comment.

5. For exact numbers such as “64 million protease sequences”, it is recommended to state the source of the data or add references to increase credibility.

This number refers to the combinatorial size of the total sequence search space for six randomized residues in TEVp (i.e. 20^6). To clarify this, we have added the term “combinatorial” to the corresponding sentence in the manuscript (line 74).

6. Please check for typos, e.g. “ubstrate” in the subsection “Recording proteolytic activity within DNA”.
Typo corrected.

7. Please add references for “Interestingly, while mutation of substrate-interacting residues was expectedly prone to activity loss...”.

The substrate-interacting residues are derived from reference 56 and are listed in Table S1, which we refer to at the end of the mentioned sentence. We now added reference 56 also to the mentioned sentence. The impact of mutating these residues is shown in Figure S9b, which we also refer to at the end of the mentioned sentence.

8. The author mentioned, “possibly due to a widened substrate pocket upon W211 replacement.” What is the basis for this conclusion? Please add references.

This hypothesis is merely based on Trp being the largest and most bulky amino acid meaning that its replacement would commonly tend to lead to more free space. Since this remains only a hypothesis in the absence of structural information, we had purposefully phrased the statement carefully (“possibly”). Additionally, we now added a note hinting towards the bulkiness of Trp (line 194-195) to clarify what we meant here and thank the reviewer for pointing this out.

9. It is recommended to add relevant descriptions for the figures to help readers understand the figure.

For example, what is the corresponding relationship between the color range changes and activity in heat maps 2b and 2c?

To clarify the color scales better, we have added a description to the figure caption. The activities displayed in Fig. 2b are the log₂ mean activities of all measured TEVp-TEVs combinations for each square (i.e. up to 20 substrates per TEVs position tested against up to 20 variants per TEVp position; max. of 400 combinations). Note that the mutational impact differs strongly between the different substrate positions (e.g. P5 can rather freely be mutated while maintaining high activity, whereas mutation of P1 is often detrimental leading to much lower activities on average). Therefore, we chose a log₂-display for this overview figure to visually resolve the entire activity spectrum better. In Fig. 2c we provide examples of the activity data underlying each square in Fig. 2b. Hence the activity scale is identical between 2b and 2c, but 2c displays individual measurements (instead of means). Since here the activity range is not as wide, log₂ transformation is not required here.

10. The author mentioned that “we framed the model as a multi-output classifier of binary activities on different substrates to account for class imbalance.” Does it mean that 135 models were trained? When training GBT, LR, SVC, and KNN models, were the embeddings of TEVp and TEVs concatenated to train a single model?

For models that are easily phrased in a multi-output setting (MLP, ESM), the substrate activities were predicted as a vector (from embeddings of TEVp), and for the other models the substrate activities were predicted as a scalar (from embeddings of TEVp concatenated to TEVs). It is hence always a single model. Details can be found in the Methods section (“Models”), where we have added two clarifying statements following the reviewer’s suggestion.

11. The distribution of mutations in the dataset has an intuitive impact on the performance of the model. It is recommended that the authors add experiments to analyze the performance of the model from different mutation distribution perspectives.

We thank the reviewer for this suggestion and point to our extensive experiments to assess the impact of different compositions of mutations in the libraries on model performance (Fig. 5e, Fig. S21). The results show that high allocation of resources to the combined library AB (equivalent to a random screen in the whole search space) leads to strong underperformance and that compositions with at least half of samples allocated to the intra-cluster libraries are favorable. These coherent empirical findings over multiple models and datasets strongly speak for the use of epistatic priors in training set design.

12. How is the test set constructed? Can the generalization ability of the model be verified? It is recommended that the authors use independent test sets of data from other proteases to verify the generalization ability of the model.

The construction of the test set is detailed in the Methods section (“Evaluation”). Briefly, the test set was randomly sampled from the available data under the constraint that samples had high quality and were measured on all substrates. The test set is hence approximately independent and identically distributed (i.i.d.) and supports the usual estimates of generalization. There were no common TEVp variants in train and test, and >80% of test set variants had a distance of at least two mutations (out of six) to the closest variant in the training set. We have added these statistics to the Methods section (“Evaluation”). Regarding generalizability of the model on other proteases, we point out that this was not the goal of our study and that no such claims are made in the manuscript. We focus task-specifically on TEVp and its substrate scope. We, however, show generalization of the MLDEEP approach with respect to performance metrics (Fig. S19), models (Fig. 5d), sample composition (Fig. 5e, Fig. S21) and choice of clusters (Fig. S20). Given that the epistasis prior should hold for all proteins, we hypothesize that the MLDEEP method (not the trained model itself) should be transferable to other protein optimization tasks. Since this is of course a hypothesis, we chose to mention it only in the discussion as a forward-looking suggestion.

13. What information does the “target activity profile” contain? Authors are advised to provide detailed information in the data preparation section.

We agree with the reviewer that this sentence was unclear and have rephrased it (lines 602-603). The exact content of the prediction target is further detailed in lines 622-626.

14. The input dimension (6 or others) of MLP should be related to the length of the input sequence. It is recommended that the authors clearly state this in the “Models”.

We have added a sentence in the Methods section “Models” to clarify that the input dimension of MLP is related to the input sequence (lines 624-625).

15. The embeddings obtained from ESM should be two-dimensional. How did the authors convert them into input for MLP?

We have added a sentence in the Methods section “Models” to clarify that the embeddings were concatenated (lines 647-648).

16. As shown in Figure 5d, MLDEEP has a significant improvement over SVC, LR, and GBT, but the improvement over KNN, ESM+MLP, and MLP is slightly weaker. It is recommended that the authors analyse this point.

We agree with the reviewer that this is indeed an interesting observation, and point out that the improvement is substantial and consistent across performance metrics (Fig. S19), models (Fig. 5d), sample composition (Figs. 5e and S21) and choice of clusters (Fig. S20). It is expected that there are variations in the extent of the effect across different types of models as their underlying learning mechanisms differ.

[end of comments]